# Understanding the Experience of Canadian Women Living with Ovarian Cancer through the Every Woman Study™

**Alicia Tone** [1,*] , **Talin Boghosian** [1] , **Alison Ross** [1] , **Elisabeth Baugh** [2] , **Alon D. Altman** [3] , **Lesa Dawson** [4] ,
**Frances Reid** [5] and **Cailey Crawford** [1]

[1] Ovarian Cancer Canada, 145 Front St E #205, Toronto, ON M5A 1E3, Canada;
boghosiantalin@gmail.com (T.B.); aross@ovariancanada.org (A.R.); ccrawford@ovariancanada.org (C.C.)

[2] School of Continuing Studies, University of Toronto, 158 St. George St., Toronto, ON M5S 2V8, Canada;
elisabeth.baugh@utoronto.ca

[3] Department of Obstetrics, Gynecology & Reproductive Sciences, Rady Faculty of Medicine,
University of Manitoba, WN5014—820 Sherbrooke St., Winnipeg, MB R3A 1R9, Canada;
aaltman@cancercare.mb.ca

[4] Department of Obstetrics and Gynecology, Faculty of Medicine, University of British Columbia,
Vancouver, BC V6Z 2K8, Canada; lesa.dawson@vch.ca

[5] World Ovarian Cancer Coalition, Toronto, ON M5A 1ES, Canada; frances@worldovariancancercoalition.org

* Correspondence: atone@ovariancanada.org

**Abstract:** The Every Woman Study™: Canadian Edition is the most comprehensive study to date exploring patient-reported experiences of ovarian cancer (OC) on a national scale. An online survey conducted in Fall 2020 included individuals diagnosed with OC in Canada, reporting responses from 557 women from 11 Canadian provinces/territories. Median age at diagnosis was 54 (11–80), 61% were diagnosed between 2016–2020, 59% were stage III/IV and all subtypes of OC were represented. Overall, 23% had a family history of OC, 75% had genetic testing and 19% reported having a *BRCA1/2* mutation. Most (87%) had symptoms prior to diagnosis. A timely diagnosis of OC ($\leq$3 months from first presentation with symptoms) was predicted by age (>50) or abdominal pain/persistent bloating as the primary symptom. Predictors of an acute diagnosis (<1 month) included region, ER/urgent care doctor as first healthcare provider or stage III/IV disease. Regional differences in genetic testing, treatments and clinical trial participation were also noted. Respondents cited substantial physical, emotional, practical and financial impacts of an OC diagnosis. Our national survey has revealed differences in the pathway to diagnosis and post-diagnostic care among Canadian women with OC, with region, initial healthcare provider, specific symptoms and age playing key roles. We have identified many opportunities to improve both clinical and supportive care of OC patients across the country.

**Keywords:** ovarian cancer; regional variation; diagnosis; treatment; psychosocial impact; genetic testing; clinical trials





## 1. Introduction

An estimated 3100 Canadian women received an ovarian cancer (OC) diagnosis—and 1950 women died from OC—in 2020 alone [1]. Standard therapy for OC consists of a combination of cytoreductive surgery and platinum-based chemotherapy. While most patients respond well to chemotherapy initially, most women with stage III/IV OC develop resistance and eventually succumb to their disease. Long-term prognosis is poor, with five-year survival rates of 41% and 20% for stage III and IV, respectively [2]. OC is not a single disease; rather, it represents several diseases that differ with respect to histology, cell and/or tissue of origin, molecular alterations, options for and response to treatment, and patient prognosis [3–8].

Ovarian Cancer Canada is the only national organization in Canada championing the health and wellbeing of women with and at risk for OC while advancing research to save lives [9]. The key pillars of our work include research, advocacy and support to women living with, or at risk of, the disease. Our ultimate vision is that women living with, or at risk of, OC can live fuller, better, longer lives. Ovarian Cancer Canada believes that everyone with OC deserves equitable access to optimal care regardless of where they live in Canada and to that end, we launched The Every Woman Study[TM]: Canadian Edition. In 2018, the World Ovarian Cancer Coalition launched the global version of this questionnaire to help highlight the challenges facing women with OC and those who care for them, and the opportunities that exist to make progress [10]. Over 1500 women participated from 44 countries, including 167 Canadians. In this study, we build on these global findings, to learn more deeply how those with OC from across Canada accessed and experienced care and identify regional or systemic opportunities for improvement. Our comprehensive national survey of >500 Canadian women living with OC explored: (1) factors that drove women to consult a healthcare provider about their symptoms; (2) factors that impacted how long it took to receive an OC diagnosis once medical attention was sought; (3) variations in pre-diagnostic and post-diagnostic care across regions; (4) the patient perspective on the impact of OC.

## 2. Materials and Methods

The global Every Woman Study[TM] [10] survey developed by the World Ovarian Cancer Coalition was adapted through consultation with individuals with lived experience with OC. Changes were made to reflect Canada's universal healthcare system and allow for open-ended questions. We also asked for the first three digits of respondents' postal codes to allow us to explore potential regional variations in responses. The final survey consisted of 110 questions covering the full continuum of clinical and supportive care for women with OC (File S1). Topics included but were not limited to: participant demographics, disease information, family history of ovarian and related cancers, OC awareness prior to diagnosis, the pathway to diagnosis, genetic testing, treatments received, clinical trials, follow-up appointments and the impacts of OC on physical/emotional wellbeing, daily life and finances. Similar to the global study, this study was not run through any institutions and external ethics approval was not sought; however, several measures were used to protect the data of those taking part (e.g., analysis of survey responses linked only to anonymous study ID, summary-level but not individual-level data presented).

The survey was available online, in both official languages (English and French), via SurveyMonkey between 28 September–6 November 2020. Only individuals diagnosed with OC who received care in Canada and were aged 18 or older at the time of the survey were eligible to participate, with no limitations on year of diagnosis. The survey was advertised through Ovarian Cancer Canada's collaborative networks, social media channels and email segments (Figure S1); participants self-selected for voluntary participation. At the end of the collection period, all responses ($n = 624$) were exported from SurveyMonkey and ineligible entries were excluded; these included repeats ($n = 42$), no questions answered ($n = 21$), diagnosis other than OC ($n = 1$) and diagnosed or treated outside of Canada ($n = 3$). The final dataset consisted of 557 responses, linked only to an anonymous study ID.

Categorical variables were summarized with counts and percentages, with Chi-squared or Fisher's Exact Test used to compare proportions between groups. Preliminary findings were shared through consultation with Canadian oncologists and the World Ovarian Cancer Coalition to identify key areas of focus for statistical analysis. To identify statistically significant predictors of key metrics (time to consulting a healthcare provider, time to diagnosis, being offered genetic testing, being offered a clinical trial), univariable and multivariable logistic regression analysis was conducted. Multinomial logistic regression analysis was conducted to identify the significant predictors associated with time to diagnosis. Statistical significance is considered as $p < 0.05$. Covariates that were significant in univariable analysis were entered into multivariable analysis, and the corresponding

results are presented. SAS version 9.4 (SAS Institute Inc., Cary, NC, USA) or R version 3.5.3 (R Foundation for Statistical Computing, Vienna, Austria) was used for conducting all analyses.

Qualitative data analysis of open-text responses was performed as follows: (1) entire data file was read to identify broad themes and phrases, with similar phrases and themes noted; (2) first phase of coding served to code and subdivide the data by the subject matter of the statements and experiences; (3) second phase of coding involved filtering responses again, assigning additional, more specific codes within each of the categories; (4) coded data was then organized into subthemes, with various categories brought together based on similarities and assigned themes.

Of note, patients with borderline tumors or who could not remember the type of ovarian cancer they were diagnosed with were excluded from all analyses based on type. Those with non-epithelial cancers were included, however, to identify unique challenges compared to women diagnosed with epithelial forms of OC. All patients were included when looking at issues related to supportive care, information needs and other qualitative assessments.

## 3. Results

### 3.1. Summary of Participants

Our final dataset consisted of responses from 557 participants (464 English, 93 French); characteristics of these participants are summarized in Table 1. Responses were received from 10 provinces and 1 territory, with the highest representation from Ontario (40%), Quebec (18%) and British Columbia (15%); the majority (86%) of respondents were from urban areas, based on postal code designations. An overview of key participant characteristics/metrics by province is provided in Table S1.

Most participants were diagnosed recently (61% within the last 5 years, 85% within the last 10 years). Median age at diagnosis was 54, ranging from 11–80 years; 63% were >50 years old and only two participants were <18 years old at the time of diagnosis several years prior to the survey. Participants were most commonly diagnosed at an advanced stage (combined 59% stage III/IV). All common types of invasive OC were represented, including high-grade serous, non-high-grade serous epithelial types (combined 23%; endometrioid, clear cell, low-grade serous, mucinous) and non-epithelial types (combined 8%; sex-cord stromal, germ cell tumors). More than half (53%) of respondents were in remission, while 34% were in active treatment for either newly diagnosed or recurrent OC at the time of the survey. A combined 41% of respondents reported either experiencing $\geq$ 1 recurrence or that their cancer had never gone away with treatment. Participants diagnosed between 2016–2020 and 2011–2015 were statistically similar with respect to important metrics such as: identity of first healthcare provider; time to diagnosis; genetic testing being offered; rates of surgery, chemo or other treatments; clinical trial being offered. In contrast, those diagnosed most recently (2016–2020) were more often stage III/IV (65% vs. 53% diagnosed 2011–2015; $p$ = 0.0166) and less often reported being in remission (43% vs. 63% diagnosed 2011–2015; $p$ = 0.0001).

Participants predominately self-identified as Caucasian (74%) or French Canadian (13%), with English (78%) or French (17%) as their first language. The majority had a post-secondary education (73%), were married (75%) and had a household income $\geq$ CAD 75,000 (63% of respondents who specified). Most (95%) participants filled out a detailed questionnaire on family history of ovarian, breast, colorectal, endometrial (uterine), pancreatic and prostate cancers on their mother's and father's side. Among all respondents, 23% had a family history of OC, either with (17%) or without (6%) a concomitant family history of breast cancer. An additional 44% of respondents had a family history of breast cancer without any reported cases of OC ("breast cancer only"). Sixty per cent of respondents had a family history of one or more related cancers, including: prostate (30%), colorectal (29%), pancreatic (16%) and uterine (10%). Furthermore, a combined 27% of participants who had genetic testing

reported a mutation in *BRCA1* (12%), *BRCA2* (7%) or in another OC gene (8%; most commonly *RAD51C/D*, *CHEK2* and the Lynch Syndrome genes).

**Table 1.** Summary of participant characteristics *.

| Variable | *n* (%) |
|:---:|:---:|
| Province or territory (*n* = 553) [1] | |
| British Columbia | 83 (15%) |
| Alberta | 70 (13%) |
| Saskatchewan | 33 (6%) |
| Ontario | 221 (40%) |
| Quebec | 100 (18%) |
| Other [2] | 46 (8%) |
| Urban vs. rural (*n* = 553) [1] | |
| Urban | 447 (86%) |
| Rural | 75 (14%) |
| Age at diagnosis (*n* = 540) | |
| ≤40 | 75 (14%) |
| 41–50 | 125 (23%) |
| >50 | 340 (63%) |
| Year of diagnosis (*n* = 545) | |
| ≤2010 | 80 (14%) |
| 2011–2015 | 131 (24%) |
| 2016–2020 | 334 (61%) |
| Time to diagnosis (*n* = 532) ** | |
| <1 month | 177 (33%) |
| 1–3 months | 160 (30%) |
| 3 months–1 year | 129 (24%) |
| >1 year | 66 (12%) |
| Type of ovarian cancer (*n* = 542) | |
| High-grade serous | 248 (46%) |
| Endometrioid | 42 (8%) |
| Clear cell | 36 (7%) |
| Low-grade serous | 36 (7%) |
| Mucinous | 11 (2%) |
| Non-epithelial cancer | 43 (8%) |
| Mixed | 6 (1%) |
| Borderline tumor | 24 (4%) |
| Other | 21 (4%) |
| Do not know or do not remember | 75 (14%) |
| Stage at diagnosis (*n* = 543) [3] | |
| I | 123 (23%) |
| II | 82 (15%) |
| III | 267 (49%) |
| IV | 54 (10%) |
| Unsure | 17 (3%) |
| Disease recurrences (*n* = 544) | |
| No | 324 (60%) |
| Has recurred at least once | 162 (30%) |
| Disease never went away | 60 (11%) |
| Current status (*n* = 545) | |
| In active treatment (newly diagnosed or recurrent) | 188 (34%) |
| In remission | 287 (53%) |
| Other | 70 (13%) |

**Table 1.** *Cont.*

| Variable | *n* (%) |
|---|---|
| **Self-reported ethnicity (*n* = 552)** | |
| Caucasian only | 410 (74%) |
| French Canadian only | 69 (13%) |
| Multiple | 22 (4%) |
| Others combined [4] | 47 (9%) |
| **Indigenous ancestry (*n* = 556)** | |
| No | 550 (99%) |
| Yes, First Nations | 3 (0.5%) |
| Yes, Métis | 3 (0.5%) |
| **First language (*n* = 556)** | |
| English | 431 (78%) |
| French | 97 (17%) |
| Other | 28 (5%) |
| **Total household income (*n* = 553)** | |
| >CAD 100,000 | 193 (35%) |
| CAD 75–99.9 K | 97 (18%) |
| <CAD 75 K | 167 (30%) |
| Prefer not to say | 96 (17%) |
| **Family history of ovarian or breast cancer (*n* = 527)** | |
| No ovarian or breast cancer | 174 (33%) |
| Ovarian cancer only | 32 (6%) |
| Ovarian and breast cancer | 88 (17%) |
| Breast cancer only | 233 (44%) |
| **Self-reported genetic testing results (*n* = 405) [5]** | |
| Mutation in *BRCA1* or *BRCA2* | 77 (19%) |
| Mutation in other gene | 34 (8%) |
| Inconclusive (variant of unknown significance) | 48 (12%) |
| Negative | 226 (56%) |
| Not sure/cannot remember/awaiting results | 20 (5%) |

[1] according to postal code designations by Canada Post. [2] includes Manitoba (*n* = 16), Newfoundland (*n* = 3), Prince Edward Island (*n* = 2), New Brunswick (*n* = 5), Nova Scotia (*n* = 19), Yukon (*n* = 1). [3] information on substage not collected. [4] includes Jewish (*n* = 8), South Asian (*n* = 8), Filipino (*n* = 5), Chinese (*n* = 3), Japanese (*n* = 3), Korean (*n* = 2), Latin American (*n* = 2), Arab (*n* = 1), Black (*n* = 0), South East Asian (*n* = 0), West Asian (*n* = 0), "other" (*n* = 15). [5] of those who had genetic testing, confirmatory data on genetic results and somatic vs. germline status of pathogenic variant not available. * information on gender identity or sexuality was not collected. ** among all respondents, including symptomatic women regardless of whether they initiated consultation about their symptoms and asymptomatic women.

### 3.2. Ovarian Cancer Awareness Prior to Diagnosis

When asked how much they knew about OC prior to their own diagnosis, the majority of participants had either "heard of it but did not know anything" (50%) or "had never heard of it" (15%); only 35% had any prior knowledge of OC (hereafter referred to as "general OC awareness") (Table S2). Participants were also asked if, prior to their diagnosis, they were aware that any of the following symptoms could be associated with OC if experienced frequently for three weeks or more: persistent bloating, abdominal pain/discomfort, urinary symptoms, difficulty eating, changes in bowel habits, menstrual irregularities, unexplained weight gain/loss, extreme/persistent fatigue. Overall, 45% of respondents had "general symptom awareness", defined as having prior knowledge of at least one OC symptom. Persistent bloating (28%), abdominal pain (27%) and menstrual irregularities (21%) were the most recognized symptoms. Only 40% of respondents had some level of 'matched symptom awareness', defined as having prior knowledge of at least one OC symptom that they themselves experienced. Both general OC awareness and general symptom awareness varied by region; respondents from Quebec (20%) and Nova Scotia (21%) had

the lowest prior knowledge of OC (range 20–56% by province) and general symptom aware-ness was lowest among Quebec respondents (32%; range 32–58% by province). Neither age nor living in an urban vs. rural setting impacted these variables. While respondents with a family history of OC had an increased level of general OC awareness compared to those with no family history of OC (48% vs. 31%; $p = 0.0006$), there was no difference in general symptom awareness between these two groups (45% vs. 44%; $p = 1.000$). Whether awareness impacted the pathway to diagnosis is explored in upcoming sections.

### 3.3. The Pathway to a Diagnosis of Ovarian Cancer

#### 3.3.1. Symptoms

Participants were asked to identify occurrence and concern level of all symptoms experienced prior to diagnosis; 87% reported at least one of the symptoms listed (Figure 1; average of three OC symptoms per respondent). Two-thirds of symptomatic respondents were either very (29%) or fairly (35%) concerned about their symptoms prior to diagnosis; younger women were more likely to be very concerned about their symptoms (35% $\leq 50$ vs. 24% > 50, $p = 0.0107$). The breakdown of symptoms experienced and considered most concerning by stage, type of OC and age at diagnosis is shown in Tables S3 and S4. Of particular note, the most common symptoms experienced were persistent bloating (60%) and abdominal pain (58%). Abdominal pain was the most concerning symptom prior to diagnosis for 28% of respondents, regardless of stage, type or age. Persistent bloating was the most concerning symptom for 18% of respondents, with an increased proportion of those diagnosed at stage III/IV or with high-grade serous cancer reporting this symptom the most troubling (22% stage III/IV vs. 13% stage I/II, $p = 0.0164$; 23% high-grade serous vs. 11% other epithelial/non-epithelial, $p = 0.0044$). While menstrual irregularities were the least commonly experienced symptom/s overall (23%), it was considered most concerning by an increased proportion of those diagnosed in stage I/II or with non-epithelial forms of OC (13% stage I/II vs. 4% stage III/IV, $p = 0.0004$; 28% non-epithelial vs. 5% epithelial, $p < 0.0001$).

A minority of respondents (11%) did not recall experiencing any symptoms prior to their diagnosis. Compared to those who did experience one or more symptoms (Table S5), asymptomatic respondents were more often >50 years old at diagnosis (76% asymptomatic vs. 62% symptomatic; $p = 0.0368$), with no differences in stage III/IV or distribution of OC types.

#### 3.3.2. Consulting a Healthcare Provider

Overall, 86% of symptomatic women initiated healthcare provider consultation about their symptoms; of these, 40% sought medical attention within 1 month, 31% between 1–3 months and 29% more than 3 months. The remaining 14% of symptomatic respondents did not initiate consultation about their symptom/s, with diagnosis occurring after seeing a medical professional for other reasons.

In comparison to respondents who did not consult a healthcare provider about their symptoms, those who sought medical attention were more likely to: be concerned about their symptoms ("very" or "fairly"; $p < 0.0001$), consider abdominal pain ($p = 0.0016$) or menstrual irregularities (0.0119) their most concerning symptom, have a higher level of education ($p = 0.0066$) and be $\leq 50$ years old ($p = 0.0037$; Figure 2 and Table S6). In contrast, those who did not consult a healthcare provider were more likely to: not be concerned about any particular symptom ($p < 0.0001$) or have a family history of breast cancer only i.e., with no concurrent family history of OC ($p = 0.0161$). Of note, almost all (96%) of those who were very or fairly concerned about their symptoms consulted a healthcare provider.

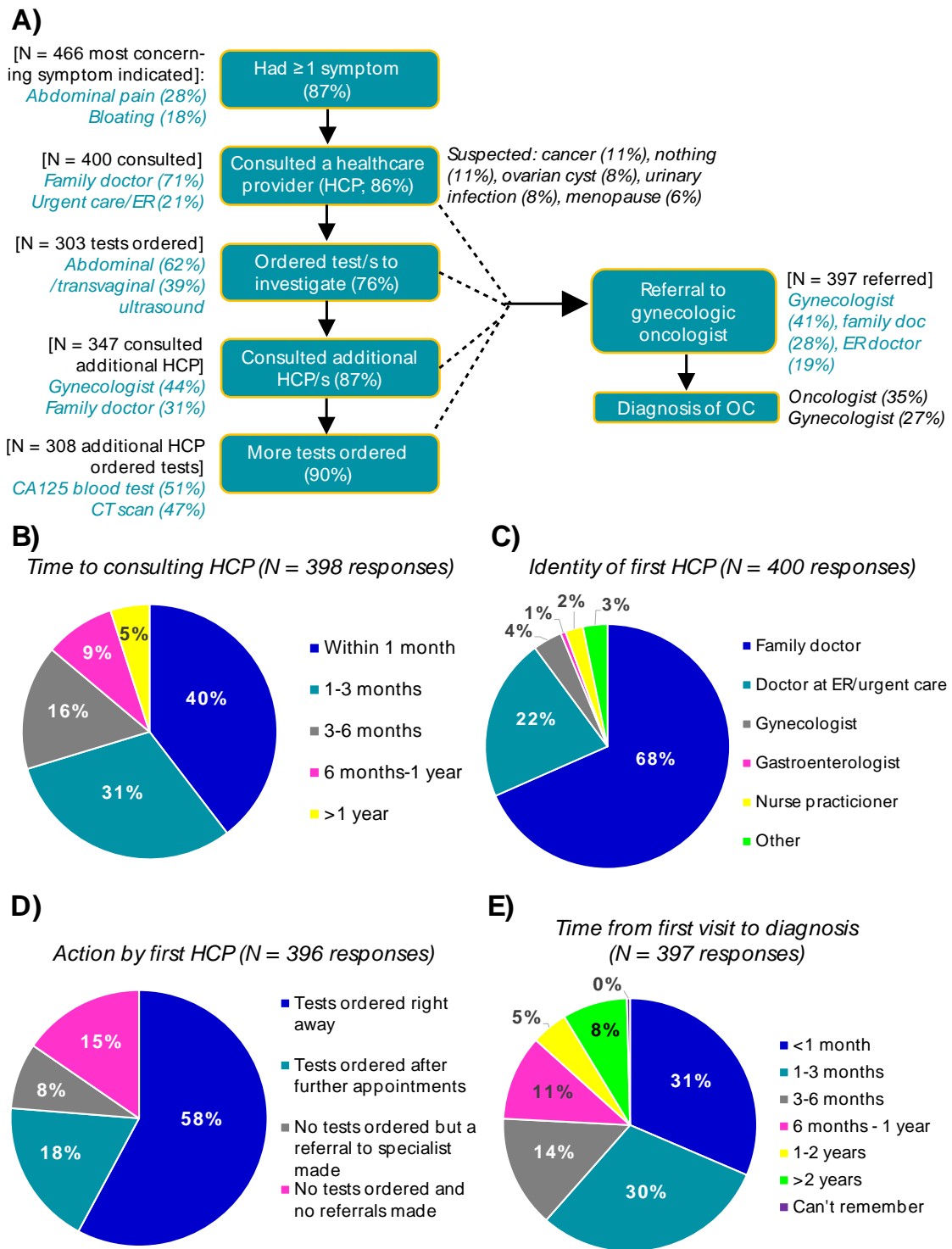

**Figure 1.** Summarizing the pathway to an OC diagnosis for symptomatic women. The pathway from initial experience of symptoms through to diagnosis is shown in panel (**A**), with the most common responses at each step highlighted. The breakdown of responses for time to consulting a healthcare provider about symptoms (**B**), identity of the first healthcare provider (**C**), action by the first healthcare provider in terms of ordering tests (**D**) and the time from first visit to OC diagnosis (**E**) are shown. Abbreviations: HCP—healthcare provider; OC—ovarian cancer.

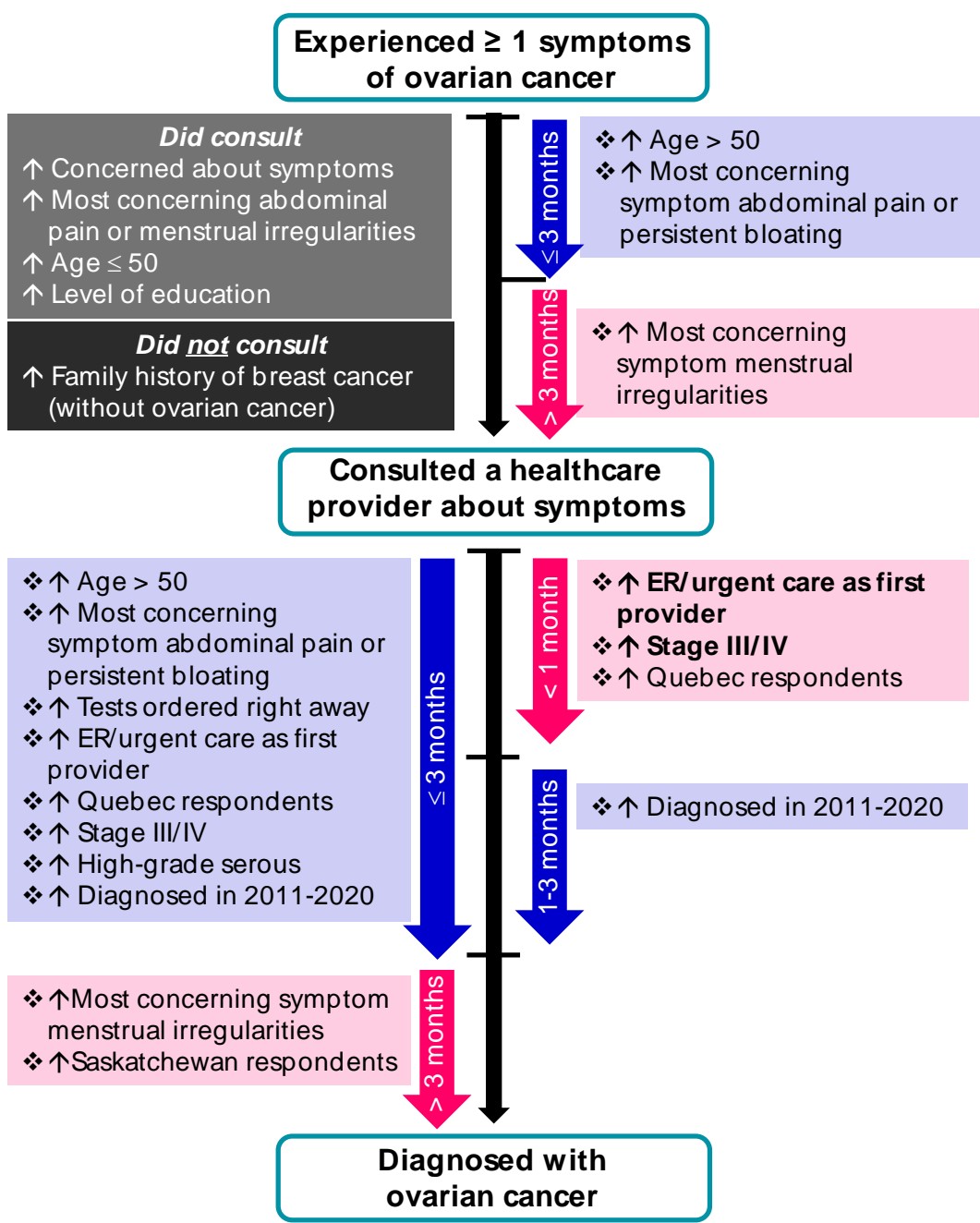

**Figure 2.** Predictors along the diagnostic pathway. Variables predicting the time between experiencing symptoms and visit with a healthcare provider (time to consult) and between the first visit and diagnosis of ovarian cancer (time to diagnosis) is shown. Women in Quebec, those who had advanced disease and those who initiated ER consultation had a shorter time to diagnosis.

However, the time to consult did not vary based on concern level. Among those who sought medical attention, univariable and multivariable logistic regression revealed that respondents who were most concerned about abdominal pain or persistent bloating were 3.53 ($p = 0.001$) and 3.02 times ($p = 0.011$) more likely to consult a healthcare provider in less than 3 months, in comparison to those most concerned about menstrual irregularities. Age greater than 50 ($p = 0.003$) also predicted a shorter time to consult ($\leq$3 months) (Table S7). There were no statistically significant differences with respect to consulting a healthcare provider, or time to consult, based on family history of OC, OC/symptom awareness, geography, language, ethnicity or income. Most participants first visited their

family doctor (71%), followed by ER/urgent care doctor (21%). The symptom profile of respondents who first presented at an ER was similar to those who first consulted their family doctor, with the top-ranked most concerning symptoms being abdominal pain (34% ER vs. 28% family doctor; $p = 0.2514$) and bloating (20% vs. 19%; $p = 0.8251$) and a similar proportion being 'very concerned' about their symptoms (25% vs. 31%; $p = 0.252$). Identity of the first healthcare provider varied by region, with family doctors being most frequent among respondents from Saskatchewan (Table S1; 80% vs. 52% in Quebec; $p = 0.0076$) and ER/urgent care doctors being most frequent among respondents from Quebec (35% vs. 7% in Saskatchewan; $p = 0.0034$). There was no difference the in identity of the first clinician for respondents living in urban vs. rural areas (data not shown). The most common suspected diagnoses at first presentation included: cancer (11%), nothing (11%), ovarian cyst (8%), urinary infection (8%) and issues related to menopause (6%).

### 3.3.3. From First Consult to Diagnosis in Symptomatic Women

Three-quarters (76%) of first healthcare providers (ER or primary care) ordered tests to investigate symptoms. Tests were ordered immediately (58%) or after further visits (18%, average of 3.1 months later). Referral to a specialist (8%) or nothing (15%) was done in the remaining cases. Abdominal ultrasound and transvaginal ultrasound were the most frequently ordered tests by the first healthcare providers (abdominal: 47% of symptomatic patients, 62% of those with tests ordered; transvaginal: 30% of symptomatic patients, 39% of those with tests ordered). ER/urgent care doctors were more likely to order a CT scan or X-ray, and less likely to order a transvaginal ultrasound, in comparison to family doctors (CT 31% ER vs. 17% family doctor, $p = 0.0159$; X-ray 21% ER vs. 11% family doctor, $p = 0.0417$; transvaginal ultrasound 26% ER vs. 41% family doctor, $p = 0.0324$).

Most symptomatic respondents (87%) went on to consult one or more additional healthcare providers prior to their diagnosis, regardless of who their first provider was. Among those who sought additional help, common responses included obstetrician/gynecologists (50%), family doctors (37%), ER/urgent care doctors (27%) and gynecologic oncologists (21%). Tests were ordered by the additional healthcare providers 89% of the time, most commonly CA125 blood test (50%), CT scan (46%) and/or abdominal ultrasound (43%). Wait times for ultrasound or other investigations after tests were ordered were not captured.

Symptomatic respondents who consulted at least one healthcare provider were referred to a gynecologic oncologist within 3 months 64% of the time, while 12% waited >1 year. Referrals to a gynecologic oncologist were most commonly sent by a gynecologist (40%), followed by family doctor (29%) or ER/urgent care (19%). Despite close to two-thirds of respondents seeing a gynecologic oncologist within 3 months of their first visit to a healthcare provider, just 34% of respondents recalled being told of their initial OC diagnosis by an oncologist; others reported being told by a gynecologist (26%), family doctor (16%), ER doctor (15%) or someone else (9%). Overall time to diagnosis from the original appointment with a healthcare provider varied, with 59% diagnosed within 3 months, 27% within 3 months–1 year and 14% in more than 1 year. Predictors of a shorter time to diagnosis (<1 month, 1–3 months or ≤3 months) are outlined in detail in Tables S8 and S9 and Figure 2.

Comparing time to diagnosis as two categories (≤3 months and >3 months; Table S8), predictors of a shorter time to diagnosis included: age > 50 ($p = 0.002$), abdominal pain ($p = 0.0034$) or persistent bloating ($p < 0.001$) as most concerning symptom, tests being ordered right away by the first healthcare provider ($p < 0.001$), ER/urgent care as first provider ($p = 0.021$), being diagnosed in Quebec ($p = 0.034$), stage III/IV disease ($p = 0.008$) or high-grade serous cancer ($p = 0.0012$). In contrast, menstrual irregularities as the most concerning symptom ($p < 0.0001$) or being diagnosed in Saskatchewan ($p = 0.049$) were predictive of a longer time to diagnosis.

Importantly, when considering three categories of time to diagnosis (<1 month, 1–3 months, >3 months; Table S9), those with the shortest time to diagnosis were more likely to first present at the ER ($p = 0.024$), have stage III/IV disease at diagnosis ($p = 0.024$)

or be diagnosed in Quebec (*p* = 0.009) compared to those diagnosed in 1–3 months. Analysis of free-text responses on women's experience leading up to an OC diagnosis (Figure 3A) revealed that those who waited >3 months to receive their diagnosis were more likely to report having a negative experience (56% >3 months vs. 24% ≤3 months; *p* <0.0001). Furthermore, respondents diagnosed in >3 months reported a lower quality of life compared to those diagnosed in 1–3 months, among respondents diagnosed with stage III disease in the two years prior to the survey (average 7/10 for >3 months vs. 7.8/10 for 1–3 months; *p* = 0.047).

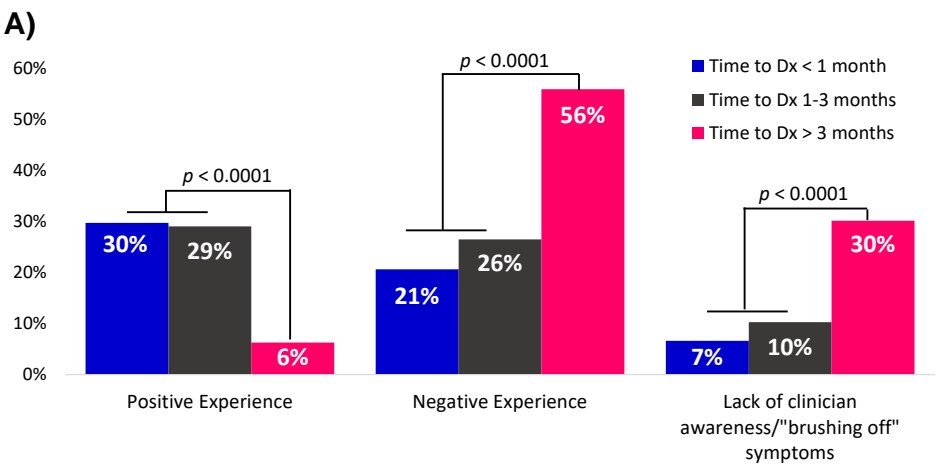

**Figure 3.** Themes revealed by respondents' experience leading up to an ovarian cancer diagnosis. Whether a respondent had an overall positive or negative experience, and the most common theme among negative experiences is shown in panel (**A**). Direct quotes from free-text responses highlighting common themes are shown in panel (**B**). Variables impacting (and impacted by) whether a respondent felt their concerns were taken seriously are shown in panel (**C**). * includes 63% of those who saw one additional HCP, 57% of those who saw two additional HCPs and 51% of those who saw ≥ 3 additional HCPs. Dx = diagnosis; HCP = healthcare provider.

Many respondents (63%) who consulted a healthcare provider about their symptoms felt that their concerns were taken seriously. However, women who were ≤50 years old ($p = 0.0099$) and were themselves very or fairly concerned about their symptoms ($p = 0.0055$) less often felt that their healthcare provider took them seriously (Figure 3C). Respondents who consulted additional providers after their first visit were also less likely to feel that their concerns had been taken seriously ($p = 0.0335$). Identity of the first healthcare provider, specific symptoms, patient OC knowledge and demographics did not play a role (data not shown). Not being taken seriously by the first healthcare provider was associated with time to diagnosis: 77% of respondents diagnosed in ≤3 months felt they were taken seriously, compared to 41% of respondents diagnosed in >3 months ($p < 0.0001$). This was supported by free-text responses, with a greater proportion of respondents diagnosed in >3 months reporting a lack of clinician awareness about OC or "brushing off" of symptoms they were experiencing (30% >3 months vs. 8% ≤3 months; $p <0.0001$).

### 3.3.4. Alternate Pathways to an Ovarian Cancer Diagnosis

Review of free-text responses from symptomatic respondents who did not initiate healthcare provider consultation about their symptoms revealed a trend towards a 'surprise diagnosis' (data not shown) following visits with their family doctor (54%) or ER/urgent care (26%). Compared to patients who sought medical help for their symptoms, these patients did not significantly differ with respect to: time to diagnosis (<3 months: 68% for did not consult vs. 59% for did consult; $p = 0.1756$); stage at diagnosis (stage III/IV: 54% vs. 63%; $p = 0.2220$); breakdown of OC types (e.g., high-grade serous: 48% vs. 45%; $p = 0.6527$).

Finally, for the minority of women who did not experience symptoms prior to diagnosis, OC was discovered while being tested or treated for something else (43%), through a 'routine' scan (14%) or examination (8%), or through other means (35%). Asymptomatic respondents were more likely to be diagnosed in <1 month, and less likely to be diagnosed in >3 months, compared to all symptomatic respondents (Table S5; <1 month: 51% vs. 31%, $p = 0.0025$; >3 months: 16% vs. 39%, $p = 0.0007$).

### *3.4. Post-Diagnostic Clinical Care*

### 3.4.1. Genetic Testing

Three-quarters (75%) of participants reported having genetic testing, predominately after diagnosis (71%) (Figure 4A). There was a great interest in genetic testing from our respondents, with only 2% declining testing. Of respondents with a family history of OC ($n = 120$), 8% had genetic testing prior to diagnosis, 79% had genetic testing after diagnosis, 10% were not offered genetic testing and 3% were not interested in testing. Corresponding proportions among those with a first-degree relative (e.g., mother, sister, daughter; $n = 31$) with OC were: 16%, 68%, 13% and 3%. The variables correlated with genetic testing after diagnosis are shown in Table S10. Factors that increased the likelihood of being offered genetic testing by univariable and/or multivariable logistic regression included: high-grade serous cancer, stage III/IV, having a family history of OC, age > 50 or being diagnosed in British Columbia. In contrast, being diagnosed in Saskatchewan or with non-epithelial forms of OC were associated with a decreased likelihood of being offered genetic testing. Of note, 94% of respondents with high-grade serous cancer, 65% with endometrioid or clear cell carcinoma and 7% with non-epithelial OC had post-diagnosis genetic testing. While those with high-grade serous cancer were tested regardless of family history, those with other types of OC were more often tested if they had a family history of OC (72% with vs. 47% without; $p = 0.0064$). All respondents treated with PARP inhibitors had post-diagnosis genetic testing performed (data not shown).

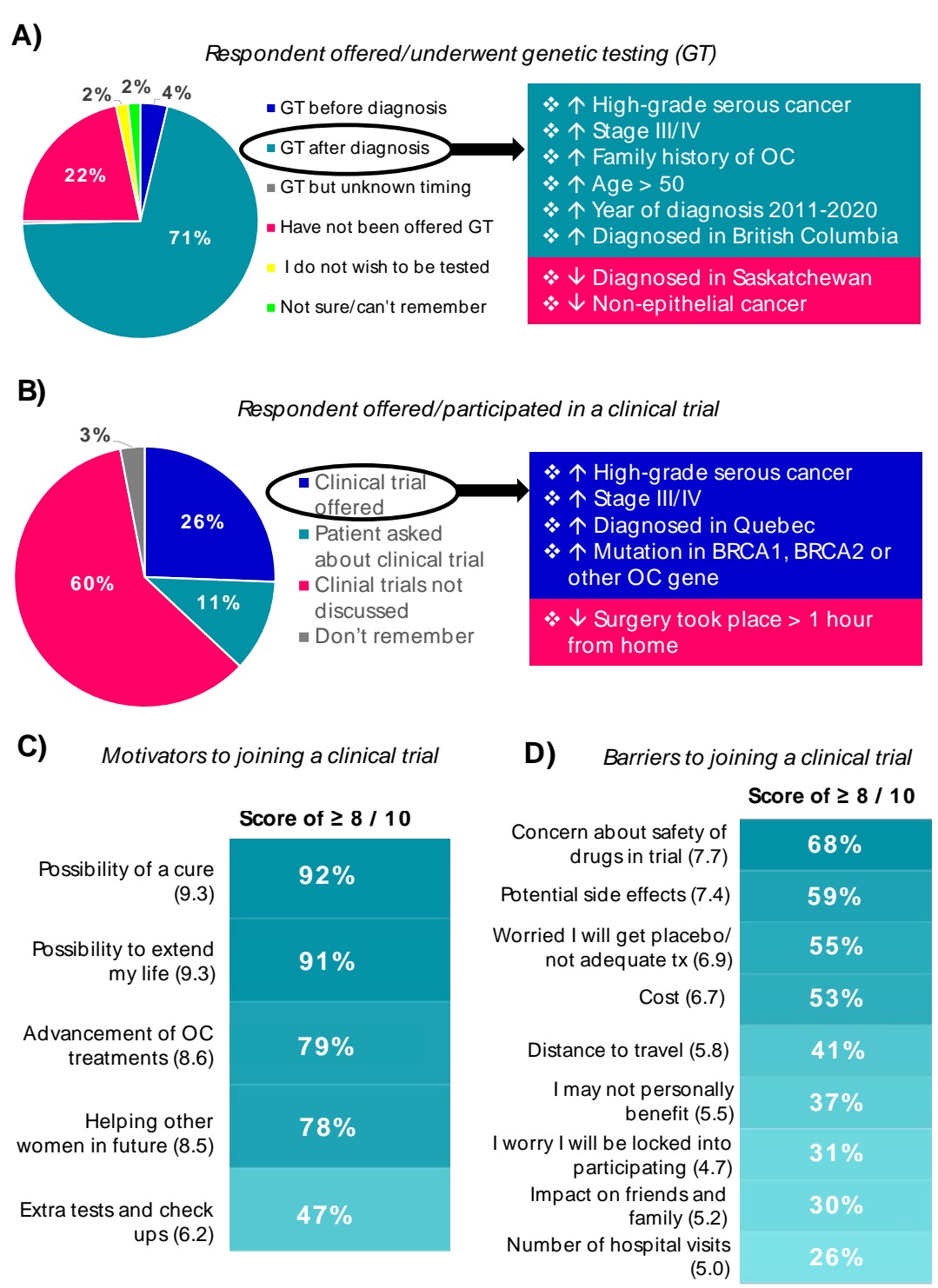

**Figure 4.** An overview of genetic testing and clinical trials. Panel (**A**) shows the breakdown of reported genetic testing before or after diagnosis. Statistically significant predictors of being offered genetic testing after diagnosis are shown at right ("↑" indicates variables that increased likelihood of being offered testing, while "↓" indicates variables that decreased likelihood of being offered testing; *p*-values in Table S10). Panel (**B**) shows the breakdown of whether a clinical trial was offered by the respondent's healthcare team, with statistically significant predictors shown at right (*p*-values in Table S12). The ranking of factors that would be considered important motivators (**C**) or barriers (**D**) to joining a clinical trial in future are shown; the proportion of respondents that gave each factor a score of 8 or more (out of 10) is indicated. Abbreviations: GT—genetic testing; OC—ovarian cancer.

A summary of self-reported genetic testing results is provided in Table 1. Of note, a greater proportion of women with vs. without a family history of OC reported a mutation in BRCA1/2 (44% vs. 10%, $p < 0.0001$); mutations in other genes were reported by the same proportion of respondents irrespective of family history of OC (9% of each). Respondents with a BRCA1 or BRCA2 mutation were almost exclusively diagnosed with high-grade serous cancer (93% vs. 88%); in contrast, multiple OC types were noted in women with mutations in Lynch Syndrome (MSH2, MSH6) or moderate risk (BRIP1, RAD51C, RAD51D) genes. Furthermore, respondents with high-grade serous cancer reported the highest rates of mutations in BRCA1/2 (26% vs. 5% of other types, $p < 0.0001$) and the lowest rates of mutations in other genes (5% vs. 19% of other types, $p < 0.0001$).

### 3.4.2. Treatments and Clinical Trials

An overview of all treatments received and variables impacting their use is provided in Table S11. Standard first-line treatment for most patients with OC includes a combination of cytoreductive surgery and platinum-based chemotherapy; consistent with this, 92% of respondents had surgery and 86% had received at least one cycle of chemotherapy. Close to half (45%) of respondents received at least one type of treatment beyond chemotherapy or surgery, most commonly PARP inhibitors (19%).

Whereas surgery did not vary based on the patient/clinical characteristics assessed, chemotherapy use was increased in high-grade serous (98% vs. 71% other types; $p < 0.0001$), stage III/IV (96% vs. 73% I/II; $p < 0.0001$) and in women >50 years old at diagnosis (93% vs. 75% ≤50; $p < 0.0001$). Of 420 women who received both chemotherapy and surgery, 73% had chemotherapy after surgery ('adjuvant'), 27% had chemotherapy prior to surgery ('neoadjuvant') and 43% had surgery and chemotherapy at different hospitals. Reported use of neoadjuvant chemotherapy was highest among respondents with high-grade serous cancer (34% vs. 14% of other types; $p = 0.0001$), stage III/IV (39% vs. 3% of I/II; $p < 0.0001$), from the province of British Columbia (50% of high-grade serous cancers vs. 29% of high-grade serous cancer from ON; $p = 0.0274$) and more recently diagnosed (32% 2016–2020 vs. 19% ≤2015; $p = 0.0049$); in contrast, there were no statistically significant differences by age.

Whether a respondent was offered a clinical trial is shown in Figure 4B; of note, 26% were offered, while an additional 11% of respondents reported asking about clinical trial enrollment. Predictors of being offered a trial by univariable and/or multivariable logistic regression (detailed in Table S12) include: high-grade serous cancer, stage III/IV, mutation in an OC risk gene or diagnosis in Quebec. In contrast, respondents who had to travel more than one hour for their initial surgery (used as a surrogate for distance to a cancer center) were less likely to be offered a clinical trial. Ultimately, 21% of patients were eligible and 18% participated in a clinical trial; only 33% of respondents recalled being provided with information on clinical trials.

When asked about the possibility of participating in a clinical trial in future, 75% of participants would be willing to travel to another hospital to take part, 16% would not want to participate at another hospital and only 10% would not want to participate at all. Respondents were not asked if they would be willing to travel out of province to participate in a clinical trial. The most important factors for deciding to join a clinical trial were "possibility of a cure" and "possibility to extend my life"; each had an average rating of >9 on a scale of 0 to 10 (Figure 4C). When asked what might prevent them from joining a clinical trial, the top-ranked reasons were "concern about safety of drugs in trial", "potential side effects" and "worried I will get placebo/not adequate treatment" (Figure 4D).

### 3.4.3. Post-Treatment Follow-Up Care

While 30% of respondents were receiving treatment at the time of the survey, the remainder had previously finished their last course of treatment: 24% ≤1 year prior, 27% between 1–5 years prior and 18% >5 years prior. Of respondents no longer receiving treatment, most (81%) were still in active follow-up and had not yet been discharged.

Follow-up appointments were most commonly every 3 months (54%) for women who finished treatment ≤1 year prior and every 6 months (40%) for those who finished 1–5 years prior (Table S13). Assessments performed at follow-up appointments included: questions about symptoms (51%), 'blood test' (49%), physical exam (46%), questions about side effects (18%), radiological assessment (14%) and questions about emotional/psychological impact (11%). All assessments, with the exception of blood tests, were less commonly reported among respondents who had been discharged >5 years prior to the survey.

### 3.5. The Many Impacts of Ovarian Cancer

We asked respondents about the many ways that OC had impacted their lives (Figure 5). Most (91%) who were no longer receiving treatment had residual long-term side effects; fatigue was considered the worst long-term effect overall, regardless of age at diagnosis. Anxiety, loss of interest in sex, menopausal symptoms and depression were more often considered among the worst long-term effects in women diagnosed at ≤50 years old, while neuropathy was more frequently cited by women diagnosed >50 years. Additional common long-term effects impacting women regardless of age included: cognition, joint pain, sleep loss and muscle pain.

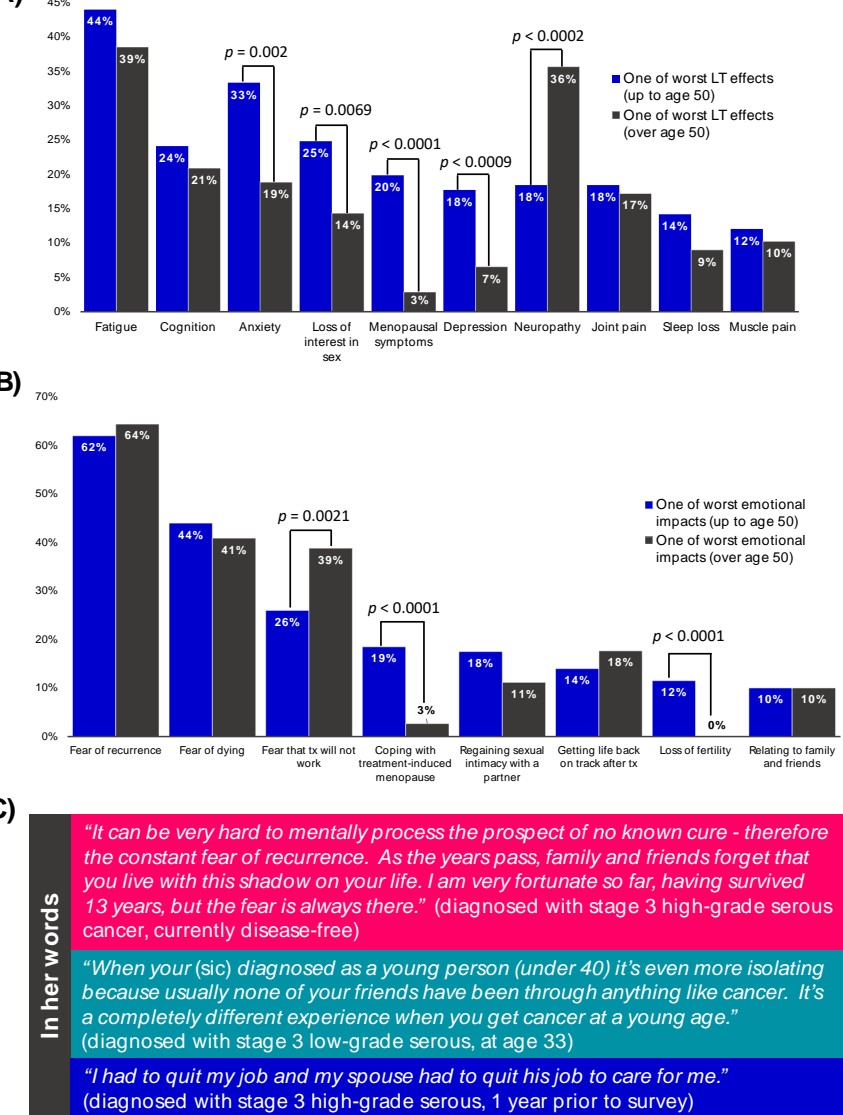

**Figure 5.** Summarizing the impact of ovarian cancer. The proportion of respondents considering a specific long-term effect of treatment (**A**) or emotional impact of ovarian cancer (**B**) one of the three

worst they experienced is shown. Statistically significant differences between those diagnosed at ≤50 years vs. >50 years old are indicated. Examples of free-text responses highlighting common themes relating the psychosocial and practical impacts of ovarian cancer are shown in panel (**C**).

The vast majority (91%) of respondents also reported experiencing emotional or psychosocial issues following their diagnosis. Fear of recurrence and fear of dying were considered the most challenging issues faced, regardless of age of diagnosis. Coping with treatment-induced menopause and loss of fertility were uniquely challenging for women diagnosed at a younger age, while fear that treatment will not work was more often cited as most challenging for women diagnosed at >50. Additional commonly cited emotional issues included: regaining sexual intimacy with a partner, getting life back on track after treatment and relating to family and friends. Respondents who had not experienced a recurrence felt in most need of emotional support at the time of diagnosis (71%), while those who did experience a recurrence felt in most need of support at the time their cancer had returned (71%) (data not shown).

The majority of respondents also reported needing one or more forms of practical support because of their diagnosis and treatment (72%) and an impact on finances (61%) (detailed in Figure S2). Of note, 27% of respondents had to personally pay for some costs related to treatment and 2% were unable to receive a specific treatment based on affordability. Quotes from free-text responses highlighting commonly cited impacts—fear of recurrence, unique issues facing younger survivors, impact on finances—are shown in Figure 5C. A summary of patient-reported information needs is included in Figure S3.

## 4. Discussion

The Every Woman Study[TM]: Canadian Edition is the most comprehensive study to date exploring patient-reported experiences of OC across most regions of Canada. Our national survey of 557 women diagnosed with OC provides a detailed and nuanced account of patient and physician actions leading up to diagnosis, post-diagnostic treatment and follow-up care, and patient perspectives on all aspects of care and the impact of OC on their physical, emotional and financial wellbeing.

It is important to note the limitations of our study. As with any retrospective patient survey, all responses are based on an individual's personal recollection, resulting in a potential for recall bias. As the survey was only open to those diagnosed with OC, questions related to the experience of symptoms prior to diagnosis could not be compared to a control group (i.e., women without an OC diagnosis); hence, discussions on symptoms are limited to whether specific symptoms or concern levels correlated to an "urgency to act" by the patient or healthcare provider. Furthermore, only patients who were still alive and feeling well enough at the time of the survey were able to participate, likely resulting in a bias towards those diagnosed more recently and/or with better prognosis disease or *BRCA1/2* mutation. Of note, the proportions of respondents who reported being "in remission" (53%) and stage III/IV (59%) in our study was similar to that reported by the global Every Woman Study[TM] (55% and 57%, respectively) [10]. Although we were able to capture perspectives from different regions across Canada, different types and stages of OC and different ages, most of our respondents were Caucasian, English or French first-language speakers, post-secondary educated, urban, married and had a high family income. Therefore, the generalizability of our findings is limited given the homogeneity of our sample. It is reasonable to conclude that those motivated to participate in this study are well-resourced and highly engaged in their healthcare, while other—particularly more marginalized patients—face barriers to this level of engagement. Although this work thoroughly examines the experiences of our participants, the population studied is not diverse enough to reliably describe the patterns of care and lived experiences of patients exposed to adverse social conditions and barriers to social resources. The profile of this study population clearly illustrates the ongoing inequities in access to and uptake in research across Canada and underestimates the relationship between an individual's social context and their pathway to OC diagnosis

and care. Concerted efforts must be made to capture and describe the experiences of all people affected by OC, particularly racialized women, women living in remote or rural communities and women from varied socioeconomic backgrounds. Such efforts would provide a more intersectional interpretation of the OC experience.

In the absence of an effective screening test for OC, timely diagnosis of OC (and a potential route for earlier diagnosis) depends on two key factors: (1) women experiencing persistent/concerning symptoms consult a healthcare provider; (2) healthcare providers respond to these concerns appropriately, through ordering of appropriate tests and/or referral to secondary care. Our finding that the majority of women reported one or more symptoms of OC prior to their diagnosis is consistent with previous studies [11]. In our study, respondents who were concerned about their symptoms, were most concerned about abdominal pain in particular, were younger than 50 years old and had a higher level of education were more likely to seek medical attention. Age greater than 50 and being most concerned about abdominal pain or persistent bloating predicted a shorter time to consult (≤3 months from symptom onset), among those who sought medical attention. Surprisingly, having a family history of OC or self-reported awareness of OC or its associated symptoms prior to diagnosis were not found to play a role in consulting a healthcare provider, or the time between experience of symptoms and seeking help (referred to as 'health-seeking interval'), in our study. Factors which may impact healthcare seeking and the health-seeking interval have been investigated in other studies, with inconsistent findings [12–18]. A Danish study [12] reported that older women (defined as 60+ years) were more likely to contact a general practitioner about gynecological 'alarm' symptoms, defined as pelvic pain, postmenopausal bleeding, or bleeding/pain during sexual intercourse. Abdominal pain has also been linked to a shorter health-seeking interval [19], similar to our study. The global Every Woman Study$^{TM}$ reported that those who "knew a lot" about OC were statistically more likely to consult a healthcare provider within 3 months of experiencing symptoms (85% vs. 75% average in full cohort) [10]; while we report a similar proportion of respondents with a high level of prior OC awareness (5%), this did not impact the health-seeking interval in our study. While some additional studies have described an association between OC symptom awareness and health-seeking interval [13,16,17], other studies have not observed an association [14,19]. In fact, a qualitative study employing focus groups and interviews in a small number of OC patients in Australia reported that several participants had intentionally ignored their symptoms due to life commitments [20]. Similarly, a higher education level has been associated both with an increased likelihood of seeking help [12] and a longer health-seeking interval [14,19]. Of note, studies in participants from low socioeconomic groups [16,18] have reported lower overall symptom knowledge and longer health-seeking interval in this group; an increase in barriers to accessing an appointment with a family physician and fearful and fatalistic beliefs about cancer contributed to a longer delay in seeking help. Of note, while respondents with a family history of breast cancer (but not OC) had comparable symptom awareness as those with a family history of OC (47% vs. 45%, respectively), the former group were over-represented among respondents who did not consult a healthcare provider about their symptoms (Table S4). Combined with the high proportion of respondents belonging to this group (44%), this suggests that more work may need to be done to raise awareness of the links between OC and breast cancer among individuals from breast cancer-only families.

We identified several factors impacting time to diagnosis once women consulted a healthcare provider about their OC symptoms. In Canada, diagnostic pathways have been developed by provincial cancer agencies; however, there are no national standards for the optimal pathway or time to OC diagnosis for women presenting with symptoms. In our study, specific symptoms, region and identity of/action by the first healthcare provider impacted a patients' time to diagnosis. Our finding that respondents from the province of Saskatchewan had a significantly longer diagnostic interval after presentation with symptoms (>3 months: 63% in Saskatchewan vs. 30% average) was unexpected, especially given the relatively small number (*n* = 33) of respondents from this region. Our finding that women presenting with menstrual irregularities as their main symptom were also more likely to wait >3 months for diagnosis suggests that the link between OC and menstrual irregularities may not be readily recognized by healthcare providers, especially in women around the age of menopause. Our finding that respondents from Quebec were more likely to experience an acute diagnosis (<1 month) may be related to the low proportion of Quebec respondents who reported family doctor as their first healthcare provider, consistent with Statistics Canada's finding that Quebec has the highest percentage of residents without a regular healthcare provider [21].

While we are unable to directly link our findings to survival outcomes, a study by the Manitoba Ovarian Cancer Outcomes (MOCO) Study Group [22] suggests that patients diagnosed in <1 month may be more likely to have a poor prognosis ("bad short") compared to those diagnosed in 1–3 months ('good short'). In the MOCO study, initial presentation to the ER was associated with worse stage-adjusted survival. In addition, 5-year survival was decreased in patients with an acute/immediate (10.8% for 7 days) compared to timely (15.6% for 76 days) diagnosis; 5-year survival subsequently decreased with increasing time to diagnosis, starting at 80 days. In contrast, a survival advantage was observed among late-stage patients diagnosed incidentally (i.e., through unrelated imaging or annual/routine physical examination in the absence of symptoms), or with a higher income within an urban setting [22]. A pooled analysis by the Ovarian Cancer Association Consortium [23] also revealed an increased risk of advanced tumor stage at diagnosis for women with a lower level of education, emphasizing the need to include individuals from all socioeconomic groups to fully capture the experience of those with OC.

In addition to a nuanced impact of a protracted diagnostic interval on survival [22], unnecessary delays in diagnosis negatively affect a patients' emotional wellbeing. In our study, recently diagnosed respondents who waited more than 3 months for their diagnosis were significantly more likely to report a negative experience during the time leading up to their diagnosis and a lower current quality of life, compared to those who waited less than 3 months. Close to one-third (30%) of those diagnosed in >3 months felt that their healthcare provider "brushed off" their concern about symptoms. This is consistent with the report by Boban et al. [20] and a similar study by Evans et al. [24] that some patients felt dismissed by general practitioners when seeking help for their symptoms. Patients with OC who had experienced a longer diagnostic delay also had decreased quality of life in a study by Robinson et al. [25]. More work therefore needs to be done on the system level to ensure that primary care physicians recognize and respond to concerning symptoms appropriately, in addition to empowering self-advocacy among women.

Genetic testing in women with OC is a crucial component of care, with a direct impact on both the patient themselves (potential for additional treatment opportunities and prevention of related cancers) and their close family members (cancer prevention); subsequently the Society of Gynecologic Oncology of Canada and others have released statements that "BRCA testing should be both routine and universal" in OC patients across Canada [26–31]. Seventy-one percent (71%) of our respondents reported having genetic testing after diagnosis, compared to 51% of respondents from the global Every Woman Study[TM] [10]. As expected, high-grade serous histology, stage III/IV disease or having a family history of OC were among the factors that were predictive of being offered testing after diagnosis. Respondents from the province of British Columbia or who were >50 years old at diagnosis were also more likely, and those from the province of Saskatchewan were less likely, to be offered genetic testing. The greater proportion of respondents >50 years being offered genetic testing (78% vs. 66% for <50) is contrary to previous studies reporting an increased uptake of genetic testing in younger cancer patients [32,33]; however, ours was not an unselected sampling of OC patients and therefore cannot be directly compared to these reports.

When asked about treatments beyond surgery and chemotherapy, 45% of respondents had received at least one additional treatment modality and 26% had been offered a clinical trial; the latter is on par with the 23.7% of global Every Woman Study[TM] participants who had been offered a clinical trial [10]. In our study, high-grade serous histology, stage III/IV disease, having a mutation in *BRCA1/2* or other OC risk gene, or being a respondent from Quebec were all predictive of being offered a clinical trial. In contrast, respondents who had to travel further for surgery were significantly less likely to be offered a clinical trial; travel time for surgery also negatively impacted use of anti-estrogens and immune therapy. While the number of respondents from rural locations was low in our study, it is noteworthy that there were trends towards a decreased likelihood of being offered a clinical trial (Table S9) or being treated with immune therapy, PARP inhibitors, anti-estrogens or radiation (Table S10), compared to urban respondents. The lack of diversity in our cohort with respect to minority ethnic populations, lower socioeconomic status and rurality precluded our ability to assess the impacts of these factors on post-diagnostic clinical care in our pan-Canadian cohort. These factors have been shown to play key roles in access to and utilization of various aspects of clinical care in previous studies, with impacts on survival [34–43]. Ovarian cancer care within other underserved communities, such as trans men (and other members of the LGBTQ2S+ community), has been understudied [44,45]; further, significant barriers to optimal care have been identified, such as discrimination by care providers and a lack of knowledge about trans-specific healthcare needs and their bodies.

A common thread throughout our findings is the identification of variations in care by region, most notably with respect to identity of the first healthcare provider, time to diagnosis and whether genetic testing or clinical trial participation was offered. The impact and underlying contributors to the potential issues highlighted in Saskatchewan (longer time to diagnosis and lower proportion of respondents with genetic testing or clinical trials offered) and Quebec (higher proportion of respondents presenting at ER, potentially linked to a "bad short" diagnostic interval) will be investigated further through focus groups with local healthcare providers and patients in the coming months. Of note, investments in local infrastructure have already been made in Saskatchewan to improve genetic testing and clinical trial participation for local patients [46].

Regardless of age at diagnosis, most women in our study experienced many physical and emotional challenges as a result of their OC diagnosis and treatment. Our finding that fatigue was the most difficult long-term physical side effect is in line with the global Every Woman Study[TM] [10]. Neuropathy, anxiety and depression were also common in both our study and the global study; however, we describe statistically significant differences by age for these effects. The most challenging emotional impact in our study, fear of recurrence, has previously been described as being prevalent in the OC patient population [47,48]. A systematic review by Ozga et al. [47] reported that the fear of cancer recurrence is a significant concern for OC patients of all ages, diagnosed with early or advanced stage disease, and throughout the continuum of care. Of note, the fear of recurrence was often increased at the end of active treatment and during follow-up appointments; this is consistent with 48% of our respondents feeling particularly in need of emotional support at the end of initial treatment, second only to the time of diagnosis (66%; data not shown). Issues related to sexual health—including loss of interest in sex, coping with treatment-induced menopause and loss of fertility—were more common in women diagnosed at a younger age in our study; several studies reveal that treatment-induced effects on sexual health are both prevalent and not adequately addressed in women with OC [49–51]. The need for additional support in all of these areas is exemplified by our finding that only 37% of respondents were offered emotional support by a healthcare provider (data not shown), slightly higher than the 28% reported by the global study [10].

## 5. Conclusions

Through the Every Woman Study[TM]: Canadian Edition, we have conducted a detailed profile of clinical care and patient-reported experiences of OC across the country. Our findings have led us to the following conclusions:

(1) The vast majority of women with OC—irrespective of age, stage or type—reported symptoms before diagnosis, yet many felt that their concerns were not taken seriously by their healthcare provider and there were wide variations in care before cancer was confirmed. Regional differences in access to care are noted and must be addressed.

(2) A family history of OC or disease/symptom awareness did not impact whether a woman consulted a healthcare provider about her symptoms, or the health-seeking interval. This demonstrates the futility of relying solely on an individuals' disease knowledge and agency to act upon symptoms appropriately. Primary care physicians also need to be armed with tools that help them recognize and respond to OC symptoms and risk factors in their patients, to ensure these women do not fall through the cracks.

(3) The contribution of hereditary causes to OC is significant, and genetic testing is essential for prevention in at-risk relatives and directed treatment for patients themselves. Despite improvements in genetic testing access for OC patients in recent years, gaps still exist (even in those with a family history of OC) and need to be resolved by systemic processes and health policies.

(4) Women with OC in Canada have a low rate of access to and enrolment in research; participation in OC clinical trials in Canada must be increased to bring new treatment options to patients who may benefit.

(5) Despite the long-lasting physical and emotional burdens of OC, support and survivorship from women after completion of treatment is an underserved area. Programs to address ongoing survivorship should be prioritized, including standardization of follow-up care when possible and strategies to manage patients' fear of recurrence.

(6) Disparities in OC care across Canada are a concern, and the homogeneity of our study sample is an important indicator that Ovarian Cancer Canada must do more to reach the many different people affected by OC. Therefore, a key strategic priority is to understand, engage, support and represent the diversity of people affected by OC. In doing so, we will access, synthesize and share research that examines the OC experiences of underrepresented groups, recognize and represent the heterogeneity of the OC community and develop resources that reflect the diversity of people we serve.

These efforts are essential as we continue to enrich our understanding of OC patient experiences and fulfill our mission of improving outcomes for all Canadians with OC.

**Supplementary Materials:** The following supporting information can be downloaded at: https://www.mdpi.com/article/10.3390/curroncol29050271/s1, File S1: Survey questions and coding; Figure S1: Recruitment materials shared through social media and Ovarian Cancer Canada's collaborative networks; Figure S2. Practical impacts of ovarian cancer; Figure S3: Patient-reported information needs; Table S1: A snapshot of regional differences (top 5); Table S2: Different types of ovarian cancer awareness prior to diagnosis; Table S3: Symptoms experienced by stage, type and age; Table S4. Most concerning symptom by stage, type and age; Table S5. Differences between symptomatic and asymptomatic respondents; Table S6: Differences between respondents who did vs. did not consult a healthcare provider about their symptoms; Table S7. Predictors of consulting a healthcare provider within three months of experiencing symptoms, among those who consulted; Table S8. Predictors of being diagnosed in ≤3 months (time to diagnosis as two categories); Table S9. Predictors of being diagnosed in <1 month and/or 1–3 months (time to diagnosis as three categories); Table S10. Predictors of being offered genetic testing after an ovarian cancer diagnosis; Table S11. Overview of treatments received and variations in use; Table S12. Predictors of being offered a clinical trial; Table S13. Summary of post-treatment follow-up care.

**Author Contributions:** Conceptualization, C.C., E.B., A.T., T.B. and F.R.; methodology, A.T. and T.B.; formal analysis, A.T. and T.B.; data curation, A.T. and T.B.; writing—original draft preparation, A.T.; writing—review and editing, A.T., T.B., A.R., E.B., A.D.A., L.D., F.R. and C.C. All authors have read and agreed to the published version of the manuscript.

**Funding:** This research was internally funded by Ovarian Cancer Canada. No external funding was obtained.

**Institutional Review Board Statement:** Ethical review and approval were waived for this study, as it was not run through any external institutions and was used to collect patient-reported data from interested participants to inform future advocacy and education needs.

**Informed Consent Statement:** Implied consent was provided by all participants, through voluntary completion of this online survey.

**Data Availability Statement:** The final coded dataset can be made available upon request.

**Acknowledgments:** The authors would like to sincerely thank all participants who took the time to fill out this survey. Thank you to the following individuals for assisting with survey creation and/or providing critical feedback on preliminary data analysis: Donna Pepin, Diane MacCormack, Kelly Anne Branco, Shaina Lee, Christa Slatnik, Jess McAlpine, Katharina Kieser, Walter Gotlieb, Annwen Jones, Clara MacKay, Robin Urquhart, Amit Oza, Marcus Bernardini and Taymaa May. Thank you to Manjula Maganti for lending her statistical expertise.

**Conflicts of Interest:** The authors declare no conflict of interest.

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
