# Peer review of "Understanding the Experience of Canadian Women Living with Ovarian Cancer through the Every Woman StudyTM"

_curroncol, doi:10.3390/curroncol29050271_

Round 1

Reviewer 1 Report

The paper by Alicia Tone and collaborators is an important work for better development of health care strategies or public health policies to improve care for women with ovarian cancer. Certainly, their findings can be source of discussion and guidance for similar studies carried out in other countries. 

Alicia Tone and co-workers drew on the global study of the World Ovarian Cancer Coalition to further study how Canadian patients experienced their own health care routines associated with the diagnosis and treatment of ovarian cancer. In this context, they obtained a detailed diagnosis showing the impressions of the women attended, regional differences in care, as well as possibilities for improving health care. Thus, they addressed issues of extreme importance to improve the treatment of a neoplasia with a high frequency of diagnoses at an advanced stage.
There are similar studies on the topic, however, the authors’ concern in the present article were to report particularities of the Canadian health system. Nevertheless, the topic of the article is relevant since the frequency of patients diagnosed at advanced stages of ovarian cancer is still too high. 
The study can provide insights on how to improve health care strategies and/or public health policies for women with ovarian cancer and, certainly, to provide guidance for similar studies carried out in other countries. 
I believe the methodology is just fine, and the authors described in DISCUSSION the main limitations of the study. 

Author Response

Thank you for your great feedback; we are happy that the importance and potential impact of our work has come across.

Reviewer 2 Report

This study evaluated the experience of ovarian cancer patients. It's novel and meaningful, and the results is reliable. Some question still should be issued.

  1. Table 1, 8% patients were non-epithelial, 4% were borderline and another 4% did not remember. Thus, those patients should not be included for analysis.
  2. Since patients with gene mutations have longer survival, this survey included those alive. Thus, the proportion of gene mutation may be elevated.
  3. Line 286, the word concerning should not be capital.

Author Response

Question 1: Table 1, 8% patients were non-epithelial, 4% were borderline and another 4% did not remember. Thus, those patients should not be included for analysis.

Response: Patients with borderline tumours or who could not remember the type of ovarian cancer they were diagnosed with were excluded from all analyses based on type. Those with non-epithelial cancers (germ cell, sex-cord stromal) were included, however, to identify unique challenges compared to women diagnosed with epithelial forms of ovarian cancer. All patients were included when looking at issues related to supportive care, information needs, and other qualitative assessments. We have added a note in our methods section (line 105-110) to clarify this.

Question 2: Since patients with gene mutations have longer survival, this survey included those alive. Thus, the proportion of gene mutation may be elevated.

Response: This is a valid point; we have added it to the limitations section of our discussion (revised line 480-481).

Question 3: Line 286, the word concerning should not be capital.

Response: Thank you for catching this; we have now corrected.

Reviewer 3 Report

The authors studied and reported "the experience of Canadian women living with OC through the Every Woman Study". They used a survey method with multi questionnaires for those recently diagnosed with OC. However, the questions are not very novel, but still, they are meant to ask a Canadian population. 

One question which I have

in line 16,

the median age at diagnosis was 54 (11-80),.... would 11-year-old patients be there in the cohort? please clarify this.

Author Response

Question: One question which I have in line 16, the median age at diagnosis was 54 (11-80),.... would 11-year-old patients be there in the cohort? please clarify this.

Response: To clarify, all participants were 18 years or older at the time they participated in the survey. The participant who reported being diagnosed at age 11 is currently 55 years old. The only other participant to be diagnosed younger than 18 is currently 27 years old. Of note, both of these participants had an unknown type of ovarian cancer (“don’t know/don’t remember” and “my doctor never told me”, respectively) so were excluded from any analyses dependent on type. A clarifying note on eligibility has been added to lines 78 and 121-122.